# A Questionnaire Study on the Use of Complementary and Alternative Veterinary Medicine for Horses in Sweden

**DOI:** 10.3390/ani11113113

**Published:** 2021-10-30

**Authors:** Karin Gilberg, Anna Bergh, Susanna Sternberg-Lewerin

**Affiliations:** 1Distriktsveterinärerna Gävle, Ludvigsbergsvägen 12, 81831 Valbo, Sweden; karingilberg@hotmail.com; 2Department of Clinical Sciences, Swedish University of Agricultural Sciences, 75007 Uppsala, Sweden; anna.bergh@slu.se; 3Department of Biomedical Sciences & Veterinary Public Health, Swedish University of Agricultural Sciences, 75007 Uppsala, Sweden

**Keywords:** CAVM, CAM, treatment, rehabilitation, therapist

## Abstract

**Simple Summary:**

As the scientific basis for most methods used in CAVM (complementary and alternative veterinary medicine) is not well-founded, there is a need for evaluation of the efficacy as well as safety of many methods. In order to better understand what evidence and knowledge are most urgently needed, we must know what methods are used, by whom and for what reasons. We asked Swedish horse owners, equine veterinary practitioners and CAVM therapists about their use of CAVM. CAVM appears to be common in Swedish horses; most horse owners used it for both prevention and treatment of injuries. The two most frequently used methods were stretching and massage. There is some collaboration between equine veterinary practitioners and CAVM therapists, which might provide an opportunity for proper diagnoses by veterinarians before CAVM therapy. Most of the study participants wanted CAVM to be more regulated, to facilitate communication and ensure animal welfare.

**Abstract:**

Complementary or alternative veterinary medicine (CAVM) includes treatment methods with limited scientific evidence. Swedish veterinarians are legally obliged to base treatments and recommendations on science or well-documented experience, but most CAVM methods are not well documented in animals. The aim of this study was to explore the use of CAVM in Swedish horses. Electronic questionnaires were distributed to horse owners, equine veterinary practitioners and CAVM therapists. Of the 204 responding horse owners, 83% contacted a veterinarian first in case of lameness, while 15% contacted a CAVM therapist. For back pain, 52% stated a CAVM therapist as their first contact and 45% a veterinarian. Only 10–15% of the respondents did not use any CAVM method for prevention or after injury. Of the 100 veterinarians who responded, more than half did not use CAVM themselves but 55% did refer to people who offer this service. Of the 124 responding CAVM therapists, 72% recommended their clients to seek veterinary advice when needed, 50% received referrals from a veterinarian, and 25% did not collaborate with a veterinarian. The two most common methods used by the respondents in all three categories were stretching and massage. Most veterinarians and therapists were not content with the current lack of CAVM regulation.

## 1. Introduction

Complementary and alternative veterinary medicine (CAVM) is defined as “... a heterogeneous group of preventive, diagnostic, and therapeutic philosophies and practices. The theoretical bases and techniques of CAVM may diverge from veterinary medicine routinely taught in veterinary medical schools or may differ from current scientific knowledge, or both.” [1]. The umbrella term includes a wide range of methods, from those that could almost be considered as conventional medicine to those where animal studies are lacking or even suggested to have no effect in animals [2,3,4,5]. The literature reviews published generally state that there is insufficient scientific research to draw any firm conclusions regarding the clinical efficacy of treatments for specific indications for many of CAVM methods [6,7,8,9,10,11,12,13,14,15,16,17,18,19,20,21]. The reasons often listed are, a limited number of studies, small sample sizes, lack of control groups, and other methodological limitations, as well as considerable heterogeneity in reported treatment effects [7,10,12]. Additionally, many studies conclude that more research is needed with an evidence-based perspective [2,3,5,6,7,8,9,12]. As more scientific evidence is gathered, the definitions of these methods may change.

Swedish legislation states that veterinarians must base their treatment methods on scientific evidence and only actors belonging to ”animal health personnel” (i.e., certified veterinary nurses, approved certified physiotherapists, approved farriers, approved certified nurses and approved certified dentists) may subject animals to injections, surgical procedures or other treatments that may cause pain or unnecessarily delayed veterinary treatments [22]. A large proportion of Swedish horses are insured, and most insurance policies cover not only costs for veterinary treatments but also CAVM treatments if performed after a referral from a veterinarian. This prerequisite presumes that veterinarians have sufficient knowledge of relevant CAVM methods and CAVM therapists to be able to refer to an appropriate person and method. As veterinarians are legally obliged to base their treatments and recommendations on science or well-documented experience, and most CAVM methods are not well-documented in animals (if at all), this presents a dilemma. On the other hand, some methods classified as CAVM are well studied in humans and may even be regarded as conventional human medicine, although animal studies are lacking. Hence, it may not be entirely obvious what methods may be regarded as evidence-based and applying the veterinary legislation is not always straightforward.

The Swedish veterinary curriculum does not include CAVM, which is similar to the situation in many other countries [23]. Most veterinarians who nevertheless practice CAVM need to obtain postgraduate education in the methods they practice [24]. Veterinary education establishments generally focus on evidence-based veterinary medicine, as science should form the basis of their education, but some consider it important to prepare their students for the increasing public interest in CAVM [23]. General knowledge of CAVM therapies is needed so that veterinarians can answer questions from animal owners.

In studies on New Zealand horse owners, collaborations between veterinarians and CAVM therapists were described as rare, and most of them did not discuss CAVM use with their veterinarian [25,26]. Similar results have been reported from human medicine, where many patients do not tell their doctor that they use complementary and alternative medicine (CAM) [27,28,29].

Little is known about why animal owners use CAVM but reports on CAM users indicate that they tend to have a higher level of education and a higher income than non-users [28,30] and that CAM is more frequently used by women than men [28]. Some patients visit a CAM therapist before they see a doctor [27,31], these people are usually younger and have a lower level of education [31] and want to avoid using prescription drugs [32]. In an Australian study, chronically ill patients were less prone to use CAM before seeking healthcare [31]. In the same study, people with a good health status frequently used CAM for prevention while people with lower income used CAM for medical problems [31].

Internationally, CAVM methods reported as commonly used by horse owners/trainers include chiropractic, physiotherapy, osteopathy, homeopathy, acupuncture [24,25,33,34,35]. Veterinarians in different countries are reported to use, or refer to therapists using, acupuncture, chiropractic, controlled walks, therapeutic shoeing, therapeutic exercises, stretching, among other methods [24,35]. There is little knowledge about the actual use of CAVM in animals in Sweden.

The aim of this study was to explore the use of and attitudes to CAVM among Swedish horse owners, Swedish equine veterinary practitioners and Swedish CAVM therapists.

## 2. Materials and Methods

The study was performed as an MSc project in the Swedish University of Agricultural Sciences [36]. Questionnaires were designed in Netigate (Netigate AB, Stockholm, www.netigate.net (accessed on 11 August 2021)) and distributed to horse owners, equine veterinary practitioners and therapists in September–October 2020. The translated questionnaires are provided as Appendix A. The questions included demographic data for the respondents, frequency, type and reasons for CAVM use, as well as money spent on CAVM (horse owners), and attitudes to the regulation of CAVM. The therapist questionnaire was constructed in collaboration with a parallel study on CAVM in dogs [37] and these results appear in both reports.

The sample was a convenience sample, where horse owners were invited to participate via social media (mainly Facebook), veterinarians were contacted via email to all members of the Swedish Veterinary Association (just over 3200 veterinarians, of which about 320 are specialized in equine medicine and surgery) and therapists were contacted via email to people listed by the Swedish national association for complementary medicine (*n* = 202) and via email to the contact points of the Scandinavian association for complementary medicine, the Swedish association for equine therapists and the Nordic association for equine masseurs. A reminder was also sent out to all email addresses (veterinarians and therapists).

The questionnaires were pilot tested on ten people well acquainted with veterinary medicine and/or animals and adapted according to comments from the test persons before launching the surveys. No data were collected from the test persons; they were only asked to assess if the questionnaire was user-friendly.

Data were extracted into Microsoft^®^ Excel (Microsoft Co., Redmond, WA, USA) spreadsheets. Descriptive statistics were calculated and univariable statistical analyses comparing the results from respondent categories of different gender and background were performed by the Chi-square test.

## 3. Results

### 3.1. Horse Owners

The questionnaire attracted 204 respondents, with 152 completing the entire questionnaire. The majority (97%) were female and most (70%) lived in rural areas. The geographical and age distribution of the respondents were fairly even, with 21% living in the north, 40% in the middle and 39% in the south of the country, most aged 18–64 years.

Most of the 204 respondents (67%) had post-high school education while 30% had attended high school. The most common equestrian disciplines that the respondents engaged in were Icelandic horses (59%), leisure riding (44%), dressage (33%), and jumping (21%). As many as 80% stated having experienced locomotor problems in their horse(s), of which 48% had recurring problems. Such problems were more commonly reported by dressage riders and less frequently by Icelandic horse riders (*p* < 0.05). Twelve percent of the horses with locomotor problems had not been seen by a veterinarian in relation to the injury, 83% of the respondents stated that contacting a veterinarian was their first choice in case of lameness and 15% that they contacted a CAVM therapist. The remaining 2% wrote that they contacted their farrier first. In the case of back pain in their horse, 52% stated a CAVM therapist as their first contact and 45% a veterinarian, the remaining 3% of the answers included chiropractic, massage and “depends on the type of back pain”. There was no association with either age or level of education as regards preferred first contact in case of lameness or back pain.

The two most commonly used methods were stretching and massage. The different methods used by the respondents for therapeutic reasons (i.e., rehabilitation after injury) or prevention (i.e., on healthy horses) is listed in Table 1.

A total of 71% of the respondents stated that the treatment helped their horse, while 4% did not think it helped and 18% were unsure. Of all respondents, only 15% did not use any CAVM method for rehabilitation after injury and 10% did not use CAVM for prevention.

There were 186 respondents to a question on side effects, most (97%) had not noticed any negative side effects from the CAVM method used while one stated that side effects had been seen but did not provide an explanation. When asked what method they believed to be effective for different problems, 177 people responded. For lameness, 32% thought stretching was effective 33% massage and 29% chiropractic, while 24% stated “none” or “other”, with many adding that this depends on the underlying cause and that a veterinarian should examine the horse first. For back problems, 71% believed massage to be effective, 56% chiropractic and 49% acupuncture, while only 5% replied “none” and 16% “other”, with some additional text stating that the underlying cause must be found first. For other disorders, such as colic, skin problems, respiratory problems, weight loss, oral problems and behavioral problems, the majority (56–70%) responded that they did not believe any of the CAVM methods to be effective.

More than half (53%) of the respondents (*n* = 186) had contacted a CAVM therapist after a recommendation from a friend, 13% based on a referral from a veterinarian and 2% via an advertisement. The remaining respondents either stated that they had not contacted a CAVM therapist (17%) or “other”, not specified (16%).

The respondents were also asked why they had chosen to contact a CAVM therapist or a veterinarian, respectively. These results are shown in Figure 1 and Figure 2.

The respondents were also given an opportunity to elaborate on why they use CAVM. These responses included statements about CAVM as a complement to veterinary care, or for preventive healthcare. Some mentioned the holistic thinking offered by the CAVM therapists, they consult CAVM therapists instead of a veterinarian when their horse has locomotor asymmetries, spinal joint hypomobility and muscular problems. Some wrote that they consult masseurs or chiropractors for themselves and want to offer their horse the same, one respondent wanted to avoid pharmaceuticals as far as possible. Reasons given for consulting a veterinarian included responses, such as trust in the veterinarian who has an education and works according to science. One mentioned consulting a veterinarian when the therapist recommends this. Many wrote that they always consult a veterinarian in case of disease or injury, as this is ethical, safe, and their responsibility as a horse owner.

When asked how much money they had spent on preventive CAVM treatments in the last year, 29% had spent up to 1000 SEK, 55.5% had spent between 1000 and 5000 SEK and 15.5% had spent >5000 SEK (*n* = 155). The corresponding proportions for CAVM treatments of injuries were 60% having spent up to 1000 SEK, 34% between 1000 and 5000 SEK and 6% having spent >5000 SEK (*n* = 156).

For products for CAVM, for prevention or treatment, 57% had spent up to 1000 SEK, 34% between 1000 and 5000 SEK and 8% spent >5000 SEK (*n* = 157).

Most (46%) of the respondents did not know if their insurance policy included non-veterinary treatments, while 28% knew this to be the case and 26% knew this was not the case. Most (59%) had not thought much about this issue, while 21% thought it important and 20% did not consider it important.

### 3.2. Veterinarians

A total of 100 veterinarians responded to the questionnaire, of which 79 responded to all questions. Seventy-five percent of the respondents were female, most lived in the southern parts of the country (43% south, 48% middle and 9% north). The majority (76%) worked in rural areas and received their veterinary education in Sweden (75%), while 17% were educated in another Nordic country and 8% in another European country. Most (69%) had more than 10 years of work experience and only 12% had less than 5 years of veterinary practice.

Only a few (9%) stated that CAVM had been part of their veterinary education. Interestingly, seven of these nine respondents were educated in Sweden and the other two *n* other European countries. Only 21% had received any postgraduate training in CAVM (*n* = 100), ten of these had attended courses/training in chiropractic treatment, six in acupuncture, four in rehabilitation, three in physiotherapy, one in homeopathy, one in osteopathy and one in general CAVM.

Most (55%), do not use any CAVM method in their work. Of those that do use CAVM, most use stretching, massage and chiropractic treatment. The same methods were most common among those who used CAVM on their own horses, but 34 of the 100 respondents stated that they did not own a horse. Products, such as cooling mud, ointment and ceramic fabrics were also used for both patients and privately owned horses, see Table 2.

Most (68%) of the respondents saw <10 equine patients with locomotor problems per week, while 28% saw 11–29 such patients per week and 4% saw >30. There was no significant difference between veterinarians with postgraduate training in CAVM and those without, with regard to the number of patients with locomotor problems.

The responses with regard to why the veterinarians who practice CAVM use these methods are illustrated in Figure 3. The number of respondents dropped for each of these questions, which might indicate either fatigue or negative reactions from some of them.

The indication for the different methods used, as stated by the respondents who use them, are shown in Table 3.

The clarifications for “other” included: shock-wave therapy, cooling and ointment for tendon injuries; controlled exercise, NSAID and shock-wave therapy for muscle injuries; shockwave, controlled exercise/physiotherapy and NSAID for ligament injuries; joint injections, school medicine first and CAVM for rehabilitation for arthritis; school medicine and CAVM for rehabilitation for neural injuries; controlled exercise/physiotherapy and NSAID for back pain. Other responses included statements that each injury must be individually assessed and properly diagnosed, referral to a skilled colleague, sometimes recommending a therapist, and not treating this type of patient.

Of the respondents, 58% (*n* = 89) stated that they have referred patients to a CAVM therapist, the most common types of therapists were massage therapists, physiotherapists, equitherapists and chiropractors. Of those who do refer to CAVM therapists, most (90%, *n* = 52) stated that they knew the educational level of the therapist, however, some were not satisfied with the level of education. Of the 71 veterinarians responding to the question about the follow-up of referrals to a therapist, 34% booked a follow-up visit with themselves or a colleague and 20% had contact with the therapist. Of the 34% who responded “other”, some wrote that they both booked a follow-up appointment and contacted the therapist Only 13% responded that they did not follow up on the treatment.

A higher proportion of female than male respondents stated that they have postgraduate training in CAVM (25% vs. 8%, *p* = 0.07), use CAVM (48% vs. 38%, *p* = 0.39), and refer to a CAVM therapist (64% vs. 43%, *p* = 0.08). There were no clear differences as regards the number of years in veterinary practice and the responses to these questions.

### 3.3. CAVM Therapists

A total of 124 therapists responded to, of which 107 completed, the questionnaire. The majority (97%) were female, 45% were 36–50 years old and 49% were 51–64 years old. Most (73%) worked in rural areas, and 33% live in the south, 54% in the middle and 13% in the north of Sweden. Only a third (31%) treated only horses, the rest worked with multiple species. The most common patients were horses, followed by people and dogs, with only a few seeing cats and other animals. The most frequently used methods include massage, stretching and acupressure, the number and proportions of therapists using different methods are shown in Table 4. Among “other” methods, kinesiotape and environmental activation were mentioned and some of the given choices were also repeated.

For 37% of 123 respondents, animal patients constitute the main source of income. The education mostly consists of courses provided by private actors, 41% of the respondents had attended such courses, while 17 had attended one or more university courses or equivalent and 7% had attended a full educational program at a university level or similar. Most (98%) were educated in Sweden, 18 of these respondents (15% of the 123) had also attended international courses and two had received their entire education outside Sweden. Most of the respondents had many years of experience with the methods they use, but 38% had practiced their methods for less than five years.

The majority (97%) stated that they are contacted directly by the animal owner, 42% via a veterinarian and 16% via other CAVM therapists (*n* = 118).

Most (72%) recommended their clients to seek veterinary advice when needed, and 50% received referrals/recommendations from a veterinarian, while 25% did not collaborate with a veterinarian (*n* = 119).

Of those who referred patients to a veterinarian, 38% booked a follow-up appointment and 33% kept in contact with the veterinarian. However, about half (55%) of the respondents responded “other” and wrote that they kept in contact with the animal owner about the progress of the treatment. One respondent stated no follow-up (*n* = 117).

The respondents’ perceptions of the methods they use are illustrated in Figure 4.

### 3.4. Legal Regulation of CAVM

The last question in all three questionnaires was if the use of CAVM in animals should be regulated to improve animal welfare and avoid mistreatments. This was a multiple-choice question allowing several options and free text was also allowed. The responses are shown in Table 5.

In the free text box, many veterinarians wrote that they want better control of the methods used and more research on what methods really work. Some veterinarians want more opportunities for postgraduate training in CAVM and some quality assured titles for such education. Many therapists wrote that they want more collaboration between veterinarians and therapists, and many wrote that they already keep records.

## 4. Discussion

The response rate/number of responses was not very high in either category, despite efforts to reach many potential respondents and keeping a neutral tone in the questionnaire, so as not to antagonize any type of respondent. It can, however, probably be assumed that most respondents are those with an interest in CAVM, therefore, selection bias cannot be ruled out. The majority were female, especially among horse owners and therapists, indicating a similar trend as studies on CAM that indicate that this is more commonly used by women [28].

The questionnaires were distributed via email to the veterinary professional association, with a known number of recipients that represent the majority of Swedish veterinarians. To reach as many horse owners as possible, social media sites were used to recruit these respondents, meaning that the number of possible respondents was unknown. The therapist questionnaire was distributed via email to various professional associations with an unknown number of end recipients, and hence the response rate cannot be calculated. A majority (85%) have stated that they use massage on animals, matching the responses of horse owners and equine veterinary practitioners. The therapist questionnaire was, however, more widely distributed within an organization for equine masseurs than in other organizations, which may have led to selection bias. All organizations received the same request to distribute the questionnaire and this skewness was beyond the control of the authors.

The horse owner questionnaire was openly distributed via social media and thus, the response rate cannot be calculated. Many equine veterinarians and CAVM therapists also own horses and could have responded to this questionnaire as well, again leading to selection bias. All questionnaires were anonymous, including IP addresses, so no check for multiple responses from the same computer could be made.

International studies indicate that many horse owners use CAVM for locomotor problems [33,35] and the results from the questionnaire to Swedish horse owners show a similar trend with many using CAVM for rehabilitation or prevention of locomotor problems [36]. However, many of the responding veterinarians stated that their main work was not on locomotor disorders, hence posing doubt about the representativeness of these respondents.

In a Swiss study, a CAVM therapist was often the first contact for horse owners who suspected locomotor problems in their horse, particularly for back pain [33]. Many of the respondents to the horse owner questionnaire in our study also stated that they first consult a CAVM therapist, especially for back pain (15% consulted a CAVM therapist first for lameness, vs. 52% for back pain). The same is reflected in other responses, where CAVM for treatment of equine back pain appears to be regarded more favorably than for lameness, by both horse owners and veterinarians. This is mirrored in international studies on CAVM for back problems in horses [24,34].

Many free text responses indicate that CAVM is frequently used by horse owners as a complement to veterinary treatments and for the prevention of health problems. This is similar to results from CAM studies [27,31] and may suggest that people extrapolate from their own experience of using CAM, to using CAVM for their animals. It is important that animal owners can distinguish preventive healthcare from the treatment of disease, as they have a responsibility to not delay veterinary treatment of an animal in need of such treatment, as stated in the Swedish animal welfare legislation. If someone, who is not categorized as animal health personnel is consulted and applies an inappropriate treatment that leads to injury or suffering of the animal, this can be subjected to legal prosecution. A majority (78%) of the CAVM therapists gave a positive response to the question about the requirement for a CAVM therapist to refer to a veterinarian, if necessary, this proportion was higher (*p* = 0.03) for the therapists than for the veterinarians’ responses to the same question (64%) and one therapist wrote that this is already the case. To ensure animal welfare and a correct diagnosis, a veterinary consultation should be the first choice, but in practice, this is not always the case.

Several therapists and veterinarians stated that they believe that there is scientific evidence that their methods are effective in animals, however, many CAVM methods lack conclusive scientific support for clinical efficacy for specific indications [6,7,8,9,10,11,12,13,14,15,16,17,18,19,20,21]. This emphasizes the importance of the initiation of well-designed research studies to ensure evidence-based information. The fact that there are withdrawal periods for many CAVM treatments before a competition, not just for pharmaceuticals, may lead horse owners to believe they are indeed effective, whereas the main objective of the withdrawal periods is to discourage competing with horses in need of any treatment, i.e., any horse that is not entirely sound. Nevertheless, many therapists also stated that they collaborate with a veterinarian and that they receive referrals from veterinarians, which should ensure that their animal patients had received the necessary veterinary treatment. However, it is not clear how the referring veterinarian can take responsibility for the CAVM treatment of the animal. Some CAVM therapists work in veterinary clinics, which means that the veterinarian is responsible for the treatment, while others operate autonomously. In the latter case, it is important that the therapists understand when a veterinary consultation is needed.

## 5. Conclusions

In summary, CAVM appears to be frequently used in Swedish horses, both for the prevention and treatment of injuries. The most common methods, according to our study, are stretching, massage and chiropractic treatment. There is some collaboration between equine veterinary practitioners and CAVM therapists but the horse owners in this study wanted more such collaboration. Closer collaborations might lead to better opportunities for proper diagnoses by veterinarians before CAVM therapy.

## Figures and Tables

**Figure 1 animals-11-03113-f001:**
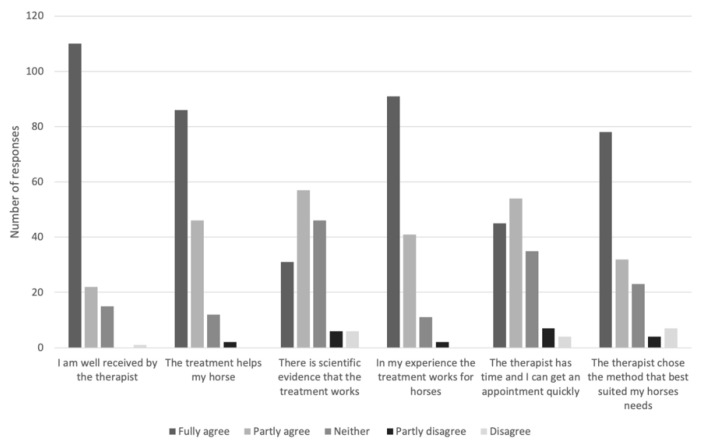
Responses by Swedish horse owners (*n* = 145) to the question “If your horse has been treated with CAVM, why did you choose that therapist and/or method?”.

**Figure 2 animals-11-03113-f002:**
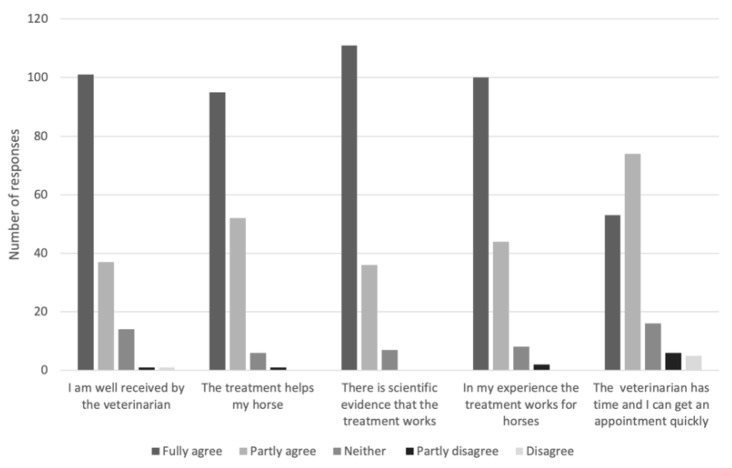
Responses by Swedish horse owners (*n* = 154) to the question “If you have consulted a veterinarian when your horse was sick or injured, why?”.

**Figure 3 animals-11-03113-f003:**
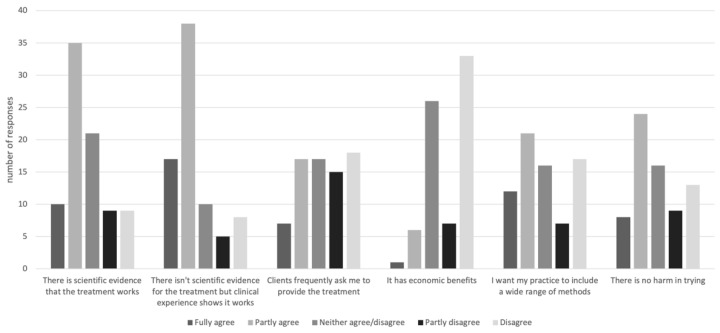
Reasons for using CAVM, as stated by Swedish equine veterinary practitioners responding to a questionnaire (*n* = 84, 78, 74, 73, 73, 70 in order shown, i.e., decreasing for each new question).

**Figure 4 animals-11-03113-f004:**
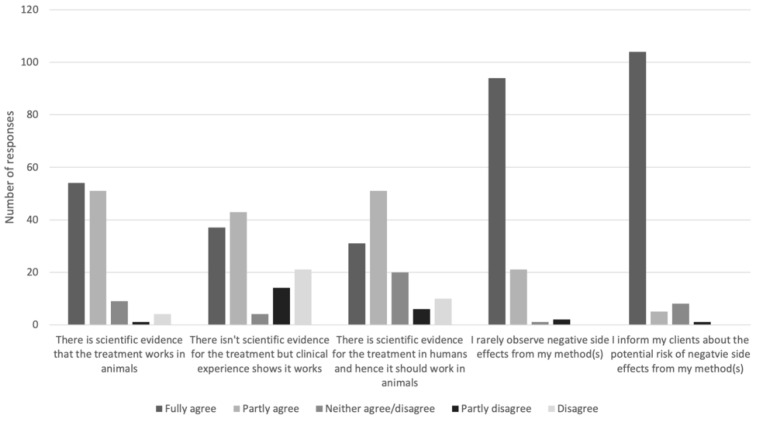
Perceptions of methods used, as reported by Swedish CAVM therapists responding to a questionnaire.

**Table 1 animals-11-03113-t001:** Frequency of use of CAVM methods used for either therapy or prevention, as stated by Swedish horse owners responding to a questionnaire (*n* = 204).

Method	>4 Times/Year	Rarely	Not at All
Acupuncture	33 (16%) *	49 (24%)	122 (60%)
Ceramic fabric	68 (33%)	27 (13%)	109 (53%)
Chiropractic	62 (25%)	65 (31%)	77 (44%)
Cooling mud	44 (22%)	36 (18%)	124 (61%)
Craniosacral therapy	6 (3%)	11 (5%)	187 (92%)
Electrotherapy/TENS/NMES	14 (7%)	32 (16%)	158 (77%)
Healing	3 (1%)	10 (5%)	191 (94%)
Homeopathy	9 (4%)	15 (7%)	180 (88%)
Hair analysis	2 (1%)	7 (3%)	195 (96%)
Iris diagnostics	1 (0.5%)	3 (1%)	200 (98%)
Kinesiology	7 (3%)	9 (4%)	188 (92%)
Laser therapy	52 (25%)	38 (17%)	114 (56%)
LED light therapy	14 (7%)	13 (6%)	177 (87%)
Magnetic therapy	11 (5%)	17 (8%)	176 (86%)
Massage	108 (53%)	55 (27%)	41 (20%)
Naprapathy	21 (10%)	35 (17%)	148 (73%)
Osteopathy	11 (5%)	20 (10%)	173 (85%)
Ointment	72 (35%)	36 (18%)	96 (47%)
Stretching	127 (62%)	36 (18%)	41 (20%)
Therapeutic ultrasound	5 (2%)	16 (8%)	183 (90%)

* Proportions indicate the distribution of responses for each method.

**Table 2 animals-11-03113-t002:** Methods used by Swedish equine veterinary practitioners responding to a questionnaire on CAVM (*n* = 100, multiple choices possible).

Method	Use for Patients	Use for Own Horse ^a^
Acupuncture	7	4
Ceramic fabric	18	18
Chiropractic	19	15
Cooling mud	18	15
Electrotherapy/TENS/NMES	8	12
Homeopathy	0	2
Kinesiology	1	0
Laser therapy	6	3
LED light therapy	1	2
Magnetic therapy	1	0
Massage	24	27
Naprapathy	1	3
Ointment	16	20
Osteopathy	2	4
Stretching	24	22
Therapeutic ultrasound	2	1
Water treadmill	14	3
Other ^b^	10	7
Do not use any CAVM	55	26

^a^ 34 respondents did not own a horse; ^b^ One respondent mentions equiband and one kinesiotape, the others give no explanation.

**Table 3 animals-11-03113-t003:** CAVM methods used or recommended for different indications by Swedish equine veterinary practitioners responding to a questionnaire. (*n* = 100, multiple choices possible).

Method	Tendon Injury	Muscle Injury	Skeletal Injury	Ligament Injury	Arthritis	Neural Injury	Back Pain
Acupuncture	5	16	4	6	10	12	12
Chiropractic	1	3	2	0	1	4	27
Electrotherapy	1	8	1	1	1	8	12
Laser therapy	41	35	10	20	14	9	6
LED light	0	0	0	0	0	0	0
Magnetic therapy	0	0	1	0	0	0	1
Massage	24	67	32	26	21	36	32
Naprapathy	0	2	1	0	2	1	6
Osteopathy	1	0	0	0	0	0	8
Stretching	21	38	22	20	14	16	15
Ultrasound	2	3	0	1	0	1	3
Water treadmill	44	56	57	38	36	43	12
None of the above	63	29	38	44	57	41	16
Other	21	12	13	14	10	16	13

**Table 4 animals-11-03113-t004:** CAVM methods used on animals by Swedish therapists responding to a questionnaire (*n* = 124, multiple choices possible).

Method		Method	
Acupressure	70 (56%)	Manipulation	23 (19%)
Acupuncture	45 (36%)	Massage	106 (85%)
Aromatherapy	1 (1%)	Mobilisation	34 (27%)
Aquatherapy/hydrotherapy	1 (1%)	Moxibustion	12 (10%)
Chiropractic	15 (12%)	Myofascial release	29 (23%)
Colloidal silver	3 (2%)	Naprapathy	8 (6%)
Craniosacral therapy	4 (3%)	Osteopathy	21 (17%)
Crystal therapy	2 (2%)	Naprapathy	2 (2%)
Distance healing	3 (2%)	Shockwave therapy	4 (3%)
Dimethyl sulfoxide (DMSO)	1 (1%)	Sound therapy	2 (2%)
Electrotherapy	13 (10%)	Stretching	72 (58%)
Healing	11 (9%)	Swimming	2 (2%)
Herbal medicine	11 (9%)	Therapeutic exercise	29 (23%)
Hirudotherapy	1 (1%)	Traditional Chinese medicine	25 (20%)
Homeopathy	7 (6%)	Trigger point therapy	44 (35%)
Infrasound	5 (4%)	Ultrasound therapy	4 (3%)
Iontophoresis	1 (1%)	Vibration therapy	8 (6%)
Kinesiology	13 (10%)	Vitamin/mineral therapy	3 (2%)
Laser therapy	56 (45%)	Water therapy	7 (6%)
LED light therapy	11 (9%)	Water treadmill	10 (8%)
Light therapy	3 (2%)	Zone therapy	3 (2%)
Ointment	19 (15%)		
Magnetic therapy	7 (6%)	Other	28 (23%)
Acupressure	70 (56%)	Manipulation	23 (19%)

**Table 5 animals-11-03113-t005:** The responses of Swedish horse owners (*n* = 156), equine veterinary practitioners (*n* = 85) and CAVM therapists (*n* = 116) to a question about regulation of CAVM in animals.

	Horse Owners	Veterinarians	Therapists
The current situation is fine, no changes needed	8 (51%)	3 (4%)	8 (7%)
A veterinary consultation should be required before CAVM treatment	48 (31%)	28 (33%)	5 (4%)
The CAVM therapist should be obliged to refer to a veterinarian if needed	112 (72%)	54 (64%)	90 (78%)
The veterinarian should be required to refer to a CAVM therapist if needed.	58 (37%)	1 (1%)	39 (34%)
Requirements on record systems for CAVM therapists should be introduced.	91 (58%)	60 (71%)	70 (60%)
Any CAVM education should include basic training in animal medicine	92 (59%)	43 (51%)	67 (58%)
Protected professional titles for CAVM therapists should be introduced	104 (67%)	34 (40%)	69 (59%)

## Data Availability

Data in coded format can be obtained from the last author.

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
