# Peer review of "A Questionnaire Study on the Use of Complementary and Alternative Veterinary Medicine for Horses in Sweden"

_animals, 2021, doi:10.3390/ani11113113_

Round 1

Reviewer 1 Report

This is an interesting paper that explores an important subject. We definitely need to increase our understanding of what makes horse owners opt for CAVM, in terms of understanding both where there are genuine benefits to CAVM and what it is that makes owners opt for CAVM even in the absence of such benefits.

This paper can certainly play a part in advancing our knowledge of the subject but needs a bit more work first.

Summary

Should “efficiency” be “efficacy”?

Abstract (and similar in Summary)

You say “Treatment methods for animals without sufficient scientific evidence are termed complementary or alternative veterinary medicine” There may be less documented use of CAVM in animals than is desirable but this is a bit of a sweeping statement that makes it sound as though there is none. This should be rephrased.

Introduction

Again, “This term includes a wide range of methods, from those that could almost be considered as conventional medicine to those where animal studies are lacking or even proven to have no effect in animals.” needs references to back this up or it appears to be simply your opinion. Likewise in Lines 52-59.

I’ve done a couple of very quick literature searches. “Osteopathy” AND “horse” gets me a couple of papers. “Massage” AND “horse” gets me a lot more papers. I think you need to do likewise and expand your literature review to say what we know about the effectiveness of CAVM and where its specific limitations are.

Likewise it would be useful to have a discussion of the limitations of proven veterinary medicine, particularly for the conditions for which owners are most likely to choose CAVM. Does this serve as a motivator for owners to choose CAVM?

Materials and Methods

Was there any strategy to the use of social media? Were particular groups used?

I saw that you do not require ethical approval but from a “good practice” point of view you should include details of how you provided an ethical process to your participants, eg informed choice to participate, right to withdraw at any point, anonymity, for how long the information would be held etc

More detail of the data analysis would be useful. I don’t feel this information currently enables replication of your study.

Results

There are a lot of per centage values quoted but it would be good to see more of a multivariate analysis. Are there any themes or correlations within your data?

Line 167-176 - “Some” and “Many” - how many? Need to be specific.

I think it would be informative to look at how many owners are using a range of CAVM practitioners and how many choose one therapy and stick to it. How does the result of this compare with those who call out the vet first? Or the education level? Or discipline?

It would have been interesting to compare the financial expenditure on CAVM with the amount spent on veterinary treatments, and whether or not they had been (perceived to be) successful. Do you have this information?

A fairly large number of vets (35? Or 35% - please make y axis clearer) in Figure 3 “partly agree” that “There is scientific evidence that the treatments work” - this contradicts your assertions earlier (and later in the  Conclusions) that there isn’t evidence. Likewise the practitioners in Fig 4. This contradiction needs proper discussion and referencing.

There is also a typo “experince -> experience.

Specific numbers needed in Lines 290-294

Masseur -> I think ‘massage therapist’ is the more normal term but maybe not in Sweden?

Fig 4 – typo “metod” -> method, “experince” -> experience

Discussion

Line 339 - I don’t think you can say that this is “ironic”. It could also suggest a high level of professionalism amongst CAVM practitioners and desire to be taken more seriously

References and more discussion needed in Lines 344-349. You can’t just say there isn’t evidence without discussing why your perception contradicts that of the vets/therapists in your study.

If you complete some multivariate analysis then I think you will find more results that are worthy of more expanded discussion.

Conclusion

Remove the last sentence - you haven’t studied the amount of scientific evidence there is for CAVM so can’t conclude that it remains a problem. On the contrary a number of vets in your study seemed to think there was evidence.

Author Response

Summary

Should “efficiency” be “efficacy”?

Answer: this could be debated, but we have changed according to suggestion. As the scientific basis for most methods used in CAVM (complementary and alternative veterinary medicine) is not well founded, there is a need for evaluation of the efficacy as well as safety of many methods.”

Abstract (and similar in Summary)

 You say “Treatment methods for animals without sufficient scientific evidence are termed complementary or alternative veterinary medicine” There may be less documented use of CAVM in animals than is desirable but this is a bit of a sweeping statement that makes it sound as though there is none. This should be rephrased.

Answer: The sentence has been rephrased to “Complementary or alternative veterinary medicine (CAVM) include treatment methods with limited scientific evidence”.

Introduction

 Again, “This term includes a wide range of methods, from those that could almost be considered as conventional medicine to those where animal studies are lacking or even proven to have no effect in animals.” needs references to back this up or it appears to be simply your opinion. Likewise in Lines 52-59.

 Answer: The following sentence has been added “Complementary and alternative veterinary medicine (CAVM) as defined in 2001 by the American Veterinary Medical Association, is “...a heterogeneous group of preventive, diagnostic, and therapeutic philosophies and practices. The theoretical bases and techniques of CAVM may diverge from veterinary medicine routinely taught in veterinary medical schools or may differ from current scientific knowledge, or both.” [1].”

I’ve done a couple of very quick literature searches. “Osteopathy” AND “horse” gets me a couple of papers. “Massage” AND “horse” gets me a lot more papers. I think you need to do likewise and expand your literature review to say what we know about the effectiveness of CAVM and where its specific limitations are.

Answer: References on literature reviews have been added. In the introduction, following paragraph has been added: “The umbrella term includes a wide range of methods, from those that could almost be considered as conventional medicine to those where animal studies are lacking or even suggested to have no effect in animals [2-5]. The literature reviews published generally state that there is insufficient scientific research to draw any firm conclusions regarding the clinical efficacy of treatments for specific indications for many of CAVM methods [6-21]. The reasons often listed are limited number of studies, small sample sizes, lack of control groups, and other methodological limitations, as well as considerable heterogeneity in reported treatment effects [7, 13-14]. Additionally, studies concludes that more research is needed based on an evidence-based perspective [2-3,5-9,12].”

In the discussion, the following was added: “Several therapists and veterinarians stated that they believe that there is scientific evidence that their methods are effective in animals, however many CAVM methods lack conclusive scientific support for clinical efficacy for specific indications [6-21]. This emphasizes the importance of initiation of well-designed research studies to ensure evidence-based information.”

Likewise it would be useful to have a discussion of the limitations of proven veterinary medicine, particularly for the conditions for which owners are most likely to choose CAVM. Does this serve as a motivator for owners to choose CAVM?

Answer: The authors are fully aware that there are methods within veterinary medicine, used in some countries, which lack level 1 evidence and might be used based on well-documented experience. However, the scope of the manuscript was not to look at veterinary medicine. Further, at least in Sweden, since animal health personnel are obliged to work according to science and well-documented experience, there are authorities that can withdraw the licence to practice if methods that are not evidence-based would be used. This is not the case with professionals outside the category animal health personnel.

Materials and Methods

Was there any strategy to the use of social media? Were particular groups used?

Answer: As stated in the material and methods, information about the survey was put on social media particularly directed to horse owners, while professionals were contacted via email to professional organisations, or directly via email to addresses listed by professional organisations. This has been spelled out more clearly in the text in M&M and Discussion.

I saw that you do not require ethical approval but from a “good practice” point of view you should include details of how you provided an ethical process to your participants, eg informed choice to participate, right to withdraw at any point, anonymity, for how long the information would be held etc

 Answer: As stated before, this type of questionnaire study does not, under Swedish law, require ethical approval and therefore no ethical board will consider assessing such studies. In the electronic questionnaire, an introductory page informed respondents about the study (content and purpose), that responding was entirely voluntary, that they would remain anonymous and that the data collected would be managed according to Swedish legislation and only presented in a format that would not allow any identification of the respondents. If the respondents chose to proceed with the questionnaire this was taken as consent. This information has been added to the manuscript.

More detail of the data analysis would be useful. I don’t feel this information currently enables replication of your study.

Answer: as only descriptive statistics and results from simple Chi square tests are presented, we do not agree, it is entirely possible to repeat this study. The sentence about data analysis has been extended to clarify what factors were included in the Chi square tests.  

Results

There are a lot of percentage values quoted but it would be good to see more of a multivariate analysis. Are there any themes or correlations within your data?

 Answer: Due to the limited number of respondents from each category of respondents and the fact that very few univariable analyses yielded any significant results we did not feel that a multivariable analysis or further data processing would be justified. In addition, as a number of respondents did not reply to all questions, a multivariable analysis would have been hampered by very few data points (respondents with incomplete data will be excluded from the analysis unless missing data are imputed which was not an option here, as too little information was available to make imputation reliable). We have presented existing correlations (and lack of correlations) in the Results. We are hesitant to proceed with further data analyses as this would entail a risk of “data torturing”.

Line 167-176 - “Some” and “Many” - how many? Need to be specific.

 Answer: Although we appreciate the comment, we disagree on this particular point as these sentences refer to voluntary free text and not responses to a direct question, The free text option was not filled out by all respondents and it is not known how many respondents would have written these things if asked straight out. Hence, even if we provide the exact number of people who wrote them, and the number of people who wrote anything in the free text box, this will not reflect the real results and instead look too precise and give a false impression of validity.

I think it would be informative to look at how many owners are using a range of CAVM practitioners and how many choose one therapy and stick to it. How does the result of this compare with those who call out the vet first? Or the education level? Or discipline?

It would have been interesting to compare the financial expenditure on CAVM with the amount spent on veterinary treatments, and whether or not they had been (perceived to be) successful. Do you have this information?

 Answer: We agree, these things would be interesting to know. Unfortunately, this cannot be assessed in our data.

A fairly large number of vets (35? Or 35% - please make y axis clearer) in Figure 3 “partly agree” that “There is scientific evidence that the treatments work” - this contradicts your assertions earlier (and later in the Conclusions) that there isn’t evidence. Likewise the practitioners in Fig 4. This contradiction needs proper discussion and referencing.

 Answer: The y-axis represents number of responses, not percentages, this has now been clarified in all figures. In the discussion, the following was added: “Several therapists and veterinarians stated that they believe that there is scientific evidence that their methods are effective in animals, however many CAVM methods lack conclusive scientific support for clinical efficacy for specific indications [6-21]. This emphasizes the importance of initiation of well-designed research studies to ensure evidence-based information.”

There is also a typo “experince -> experience.

 Answer: Corrected, thanks for spotting this

Specific numbers needed in Lines 290-294

 Answer: this may be caused by the picture of the figure obscuring the line numbers. Due to the format of the template we are unable to crop the figure from the right side.

Masseur -> I think ‘massage therapist’ is the more normal term but maybe not in Sweden?

Answer: Change has been performed according to suggestion. Of the respondents, 58% (n=89) stated that they have referred patients to a CAVM therapist, the most common types of therapists were massage therapist, physiotherapist, equitherapist and chiropractor.”

Fig 4 – typo “metod” -> method, “experince” -> experience

 Answer: corrected, thank you

Discussion

Line 339 - I don’t think you can say that this is “ironic”. It could also suggest a high level of professionalism amongst CAVM practitioners and desire to be taken more seriously

 Answer: The word “ironically” has been deleted.A majority (78%) of the CAVM therapists gave a positive response to the question about requirement for a CAVM therapist to refer to a veterinarian if necessary, this proportion was higher (p=0.03) for the therapists than for the veterinarians’ responses to the same question (64%) and one therapist wrote that this is already the case.”

References and more discussion needed in Lines 344-349. You can’t just say there isn’t evidence without discussing why your perception contradicts that of the vets/therapists in your study.

Answer: This has been elaborated on in other parts of the manuscript. In the Discussion the following is now written: Several therapists and veterinarians stated that they believe that there is scientific evidence that their methods are effective in animals, however many CAVM methods lack conclusive scientific support for clinical efficacy for specific indications [6-21]. This emphasizes the importance of initiation of well-designed research studies to ensure evidence-based information.

If you complete some multivariate analysis then I think you will find more results that are worthy of more expanded discussion.

 Answer: We assume you mean multivariable and not multivariate. A lack of significant results in univariable analyses will not result in significant results in multivariable analyses. Se response above.

Conclusion

Remove the last sentence - you haven’t studied the amount of scientific evidence there is for CAVM so can’t conclude that it remains a problem. On the contrary a number of vets in your study seemed to think there was evidence.

Answer: Changes have been performed according to suggestion.

Additional information for the reviewers and editors. An international group, with representation among the authors of the present manuscript, has the last year conducted systematic literature reviews on CAVM. The reviews include methods such as electrotherapy, light therapy, acupuncture, manipulation therapies, soft tissue mobilisation, as well as therapies outside these groups. Over 15 000 abstracts, looking at the scientific documentation regarding the clinical efficacy of these methods for the species horse, dog and cat, have been reviewed. In total, approximately 200 full-length articles have been included in the reviews, after being assessed using a standardised assessment protocol regarding the studies risk of bias. The first of several manuscripts has been published and the rest will be submitted during this fall.

The scope of the present manuscript was to investigate the use of CAVM in Sweden. It gives information on the area and highlights the importance to distinguish between having an opinion and having scientific evidence to back up that opinion. Ie just because some veterinarians think there are sufficient scientific evidence to support the clinical efficacy of a specific method- is not a guarantee that there actually is.  

Reviewer 2 Report

This manuscript describes the views of Swedish veterinarians, horse owners and complimentary/alternative therapists on the subject of complimentary/alternative therapy. This is an important topic given both the potential benefits from good therapy but also the potential for negative welfare if an ineffective modality is chosen and/or veterinary treatment is delayed.

The manuscript is clear and easy to read. Whilst there is some overlap with other studies, repeatability of these types of studies in different countries is important.

Minor corrections

Line 28/29 suggest make it clear that over half the veterinarians did not use CAVM themselves but 55% did refer to people who offer this service

I was slightly confused regarding physiotherapy. It was mentioned in the introduction and briefly elsewhere but did not appear to be an option throughout the survey. This may be a difference in the term between countries. In the U.K. a physiotherapist is usually a human physiotherapist  who has then undertaken a masters degree in veterinary physiotherapy. They work using evidence based treatments and are highly qualified. This would be very different to a chiropractor/massage therapist/someone recommending stretches. I think some clarity on why the term physiotherapy was left out or if it was included under as part of a different heading would be beneficial

line 299 - I agree most responders would have had an interest in CAVM and so suggest specifically stating selection bias is likely to have occurred (as is always the case with these studies)

References - appear to be in a random order?? 

Author Response

 Minor corrections

Line 28/29 suggest make it clear that over half the veterinarians did not use CAVM themselves but 55% did refer to people who offer this service

Answer: Changed according to suggestion.” Of the 100 veterinarians, who responded, over half did not use CAVM themselves but 55% did refer to people who offer this service.”

I was slightly confused regarding physiotherapy. It was mentioned in the introduction and briefly elsewhere but did not appear to be an option throughout the survey. This may be a difference in the term between countries. In the U.K. a physiotherapist is usually a human physiotherapist  who has then undertaken a masters degree in veterinary physiotherapy. They work using evidence based treatments and are highly qualified. This would be very different to a chiropractor/massage therapist/someone recommending stretches. I think some clarity on why the term physiotherapy was left out or if it was included under as part of a different heading would be beneficial.

Answer: If we understand the comment correctly, the question was why physiotherapy was left out in the pre-set answers regarding different types of therapies. The short and simple answer is that the methods listed are “specific methods” and not collective names/umbrella terms that include a variety of methods, such as physiotherapy- were a wide range of methods are used. Thus, physiotherapy doesn´t represent one specific therapy, but a way of making a functional diagnosis and selecting optimal treatments using the physiotherapy process. As an example, one can look at the literature review that is published on animal physiotherapy, demonstrating the different methods included in review (Mahaseth PK, S. Raghul. Veterinary physiotherapy - a literature review. International Journal of Science & Healthcare Research. 2021; 6(1): 288-294.).

In addition, animal physiotherapist is not a protected professional title in Sweden (while it is a protected title in human medicine) and using the title in the questionnaires would have led to uncertainties.

line 299 - I agree most responders would have had an interest in CAVM and so suggest specifically stating selection bias is likely to have occurred (as is always the case with these studies)

 Answer: Changed according to suggestion.It can, however, probably be assumed that most respondents are those with an interest in CAVM, therefore a selection bias cannot be ruled out.”

References - appear to be in a random order?? 

Answer: Correct, this has slipped through our own revision- thank you for making us aware of the fault. It has now been corrected.

Round 2

Reviewer 1 Report

Thank you for the revisions. This is now fine.